# Crucible: Quantifying the Potential of Control Algorithms through LLM Agents

**Lianchen Jia[1], Chaoyang Li[1], Houde Qian[1], Tianchi Huang[1], Jiangchuan Liu[2], Lifeng Sun[1,3]***
[1]Department of Computer Science and Technology, Tsinghua University,
[2]Simon Fraser University, [3]BNRist
jlc21@mails.tsinghua.edu.cn, sunlf@tsinghua.edu.cn

## Abstract

Control algorithms in production environments typically require domain experts to tune their parameters and logic for specific scenarios. However, existing research predominantly focuses on algorithmic performance under ideal or default configurations, overlooking the critical aspect of Tuning Potential. To bridge this gap, we introduce `Crucible`, an agent that employs an LLM-driven, multi-level expert simulation to turn algorithms and defines a formalized metric to quantitatively evaluate their Tuning Potential. We demonstrate `Crucible`'s effectiveness across a wide spectrum of case studies, from classic control tasks to complex computer systems, and validate its findings in a real-world deployment. Our experimental results reveal that `Crucible` systematically quantifies the tunable space across different algorithms. Furthermore, `Crucible` provides a new dimension for algorithm analysis and design, which ultimately leads to performance improvements. Our code is available at https://github.com/thu-media/Crucible.

## 1 Introduction

Control algorithms are the core mechanisms that dynamically regulate system behavior to achieve specific objectives, with widespread applications from industrial automation to complex computer systems [1, 2]. These algorithms play pivotal roles in various contexts, including gait control in robotic systems [3], motion control for autonomous vehicles [4], adaptive bitrate (ABR) control [5–8] and congestion control [7, 9] in network applications, and scheduling control within data centers [10]. Such control algorithms are essential for ensuring system stability while simultaneously optimizing overall performance.

However, current research predominantly evaluates algorithms based on their performance under ideal conditions or with default parameters [11, 12]. This paradigm overlooks a critical reality of production environments: algorithms are always tuned by domain experts to adapt to specific scenarios. The performance of an algorithm in the wild is therefore not just a function of its default design, but also of its inherent adaptability—a property we define as its **Tuning Potential**. The lack of a systematic way to measure Tuning Potential makes it difficult to compare the practical adaptability of different algorithms and hinders its establishment as an explicit design goal.

Evaluating Tuning Potential presents a significant challenge that extends beyond conventional parameter sensitivity analysis [13]. The assessment must encompass deeper, logic-level modifications, including the incorporation of additional control branches or the integration of novel components—interventions commonly employed by domain experts. The efficacy of such structural adjustments is linked to expert subjective understanding of the underlying algorithmic logic [14].

---

*Corresponding author.

39th Conference on Neural Information Processing Systems (NeurIPS 2025).

This complex interplay between objective performance metrics and subjective understanding factors impedes the establishment of a Tuning Potential evaluation framework.

To address this gap, we introduce `Crucible`, the first framework designed to quantitatively evaluate the Tuning Potential of control algorithms. `Crucible` operates on two key principles. First, it employs a Large Language Model (LLM)-driven, multi-level expert simulation agent that is defined with varying capabilities, such as the ability to utilize optimization tools or perform multi-step reflection, thereby emulating how developers with different expertise levels approach algorithm tuning. This approach circumvents the prohibitive costs associated with large-scale subjective studies [15]. Second, it establishes a formalized metric to normalize potential scores across diverse environments. This metric characterizes each task environment by analyzing the performance profile that emerges when a set of probe algorithms is applied to it, thereby enabling consistent comparisons across different domains.

To validate the effectiveness and generalizability of the `Crucible` framework, our evaluation encompasses a wide spectrum of case studies, from classic control tasks (Cart-Pole [16]) to complex computer systems (ABR and scheduling control), and we validate our findings in a real-world deployment. Our experiments demonstrate that `Crucible` consistently identifies a larger optimization space than traditional Bayesian methods. By analyzing the results, we identify that an algorithm's representational capacity and comprehensibility are two primary factors influencing its potential. Finally, we show that these insights can guide targeted algorithm redesign, leading to significant performance improvements.

Our key contributions are summarized as follows:

- We identify and formalize Tuning Potential as a critical, yet overlooked, dimension in algorithm evaluation. We argue that neglecting this dimension limits the practical impact of algorithms and hinders its adoption as a core design objective (Section 2).

- We propose `Crucible`, the first system designed to quantify the Tuning Potential of control algorithms. Through an LLM-based multi-level expert simulation agent and a formalized environmental metric, it provides a systematic, quantifiable standard for measuring algorithmic adaptability (Section 3).

- We validate `Crucible`'s effectiveness and generalizability across a diverse range of scenarios, from classic control tasks to complex computer systems, including real-world validation. Our results show that `Crucible`'s quantitative analysis offers clear guidance for algorithm design, ultimately enhancing both performance limits and practical value (Section 4).

## 2 Motivation

### 2.1 LLM-Based Human Behavior Simulation

LLMs, trained on internet-scale corpora, have demonstrated exceptional performance not only in traditional natural language processing tasks such as translation, summarization, and question answering [17, 18], but also exhibited emergent general knowledge and analytical reasoning capabilities in accordance with scaling laws [19, 20]. LLM advancements have prompted researchers to explore the application of LLMs in human behavior simulation. In social sciences, numerous studies [21–23] have investigated LLM-based user behavior simulation and subjective perception assessment, successfully generating credible individual behavioral patterns and their resulting emergent social behaviors within groups. In the field of recommendation systems, researchers [24–26] have effectively utilized LLMs to simulate diverse user behaviors, including browsing, searching, and content consumption activities. In our research, we apply LLMs to simulate developers' understanding and adjustment processes of algorithms, circumventing the high costs and time expenditures associated with personnel training in traditional large-scale subjective studies, thereby providing an efficient and scalable new method for algorithm evaluation.

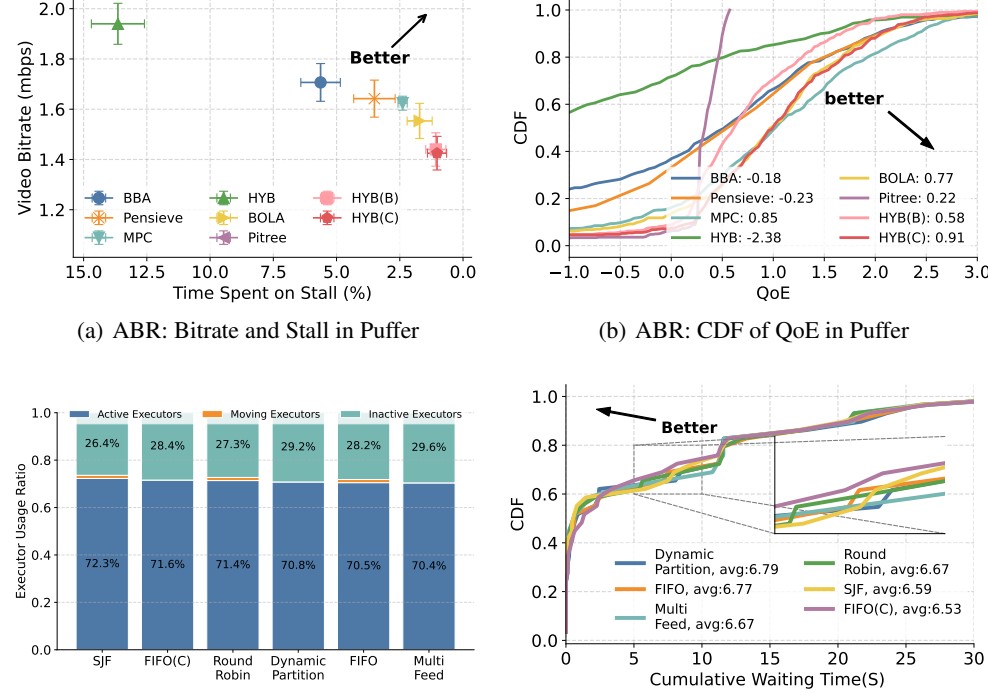

(a) ABR: Bitrate and Stall in Puffer

(b) ABR: CDF of QoE in Puffer

(c) Scheduling Control: Executor utilization Ratio in 2G Input Sizes

(d) Scheduling Control: Cumulative Waiting Time in 2G Input Sizes

Figure 1: The Importance of Algorithm Potential in the Real World

## 2.2 Control Algorithms in the Real World

We selected two representative examples from the field of computer systems—ABR control in network transmission and scheduling control in distributed systems—to demonstrate the effectiveness of `Crucible` in real-world control algorithms.

**Adaptive Bitrate Control** Adaptive bitrate algorithms enhance the quality of experience (QoE) by dynamically selecting the bitrate for the next playback chunk, aiming to improve playback quality while preventing stalling events [5]. Common control approaches include buffer-based algorithms (BBA [27], BOLA [28]), hybrid methods combining bandwidth and buffer information (MPC [29], HYB [30]), decision tree-based approaches (Pitree [31]), and reinforcement learning (RL)-based solutions such as Pensieve [32].

**Scheduling Control** Distributed directed acyclic graph (DAG) task scheduling algorithms are responsible for efficiently allocating and executing interdependent tasks in distributed computing environments, where dependencies are represented through DAGs [10]. These scheduling algorithms optimize task allocation mechanisms while considering resource utilization and task dependencies to enhance overall computational efficiency and reduce completion time. Common scheduling strategies include Shortest Job First (SJF), Shortest Remaining Time First (SRTF), Fair Scheduling, First-In-First-Out (FIFO), Round Robin, Dynamic Partitioning [33], Multi-feed methods [34], and Tetris [35].

## 2.3 From Algorithm Design to Production Deployment

During the algorithm design phase, researchers typically focus on theoretical performance [11] and cross-scenario robustness [12], pursuing universal performance across a wide range of application scenarios. However, when algorithms are actually deployed in production environments, we can obtain stable feature information specific to the application scenario, which enables targeted optimization. For instance, user bandwidth demands in network content provider services [36] and system loads in task scheduling scenarios [37] often exhibit stable and predictable patterns of change. This stability and predictability of scenario characteristics allow developers to customize optimization strategies

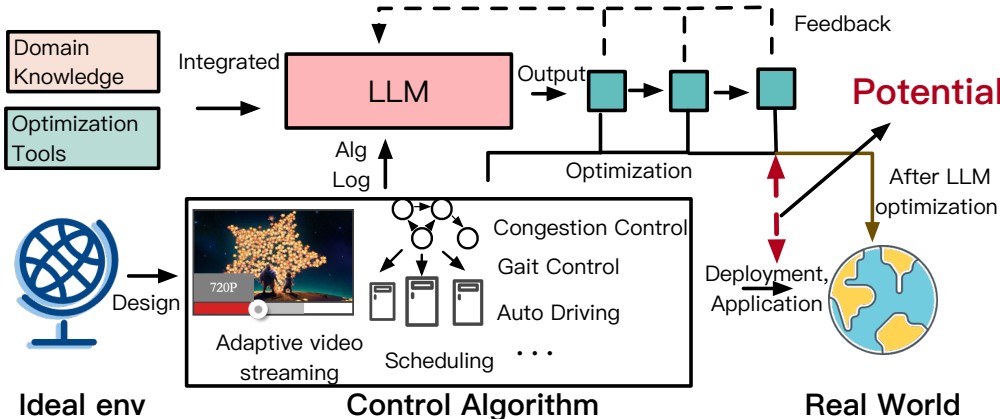

Figure 2: `Crucible` System

according to specific application environments, thereby achieving performance that surpasses general-purpose algorithms in practical deployments.

## 2.4 Experimental Validation of the Importance of Potential

Figure 1 demonstrates our experimental results in the domains of ABR and scheduling (detailed experimental setup in Appendix A), validating the importance of tunability. Our findings reveal: 1) Algorithms based on simple logic, after appropriate tuning, can significantly outperform complex designs. In the ABR task, as shown in Figure 1(a), HYB(B) optimized through Bayesian optimization [38] and HYB(C) adjusted via `Crucible` reduced video playback stalling time by 92%. Figure 1(b) further indicates that these optimizations elevated the overall QoE ranking from the lowest position to fourth and first place, respectively. Similarly, in the scheduling problem, the simple FIFO algorithm tuned by `Crucible` achieves sub-optimal executor utilization as Figure 1(c) and the optimal cumulative waiting time as Figure 1(d). 2) Effective tuning encompasses not only parameter optimization but also improvements in logical structure. In the ABR task, while parameter-optimized HYB(B) still could not surpass the QoE performance of BOLA and RobustMPC, HYB(C) with logical adjustments through `Crucible` ultimately achieves the best QoE performance, thoroughly demonstrating the importance of algorithm logic adjustment.

## 3 Design

The overall design of `Crucible` can be seen in Figure 2. We introduce the design of `Crucible` from three aspects: Section 3.1 describes the workflow of the `Crucible` agent, Section 3.2 provides a formalized definition of potential, and Section 3.3 explains the interaction between `Crucible` and the control algorithms being evaluated.

### 3.1 Agent Workflow

**Domain Knowledge Acquisition** We inject domain knowledge into LLMs through system prompts across three dimensions: task description, optimization objectives, and environment overview. The task description defines the basic information of the control task, including the input states and the scope of optional output behaviors; the optimization objectives specify the improvement direction and evaluation criteria for the control algorithm; the environment overview provides key characteristics and constraints of the testing scenarios. These three dimensions construct a comprehensive knowledge framework, enabling LLMs to thoroughly understand the task context, optimization direction, and operational environmental characteristics.

**Tool Utilization** The potential of algorithms is primarily manifested in two key dimensions: first, the representational capacity of the algorithm, referring to the breadth and granularity of its control space—algorithms with higher control dimensions and greater precision exhibit richer performance variability in the hyperparameter space, thus possessing greater optimization potential [39, 40]; second, the comprehensibility [41, 42] of the algorithm—algorithms with higher structural transparency

enable developers to implement targeted functional enhancements through logical reconstruction. The former can be systematically improved through mature automatic optimization techniques, while the latter relies on the abstract understanding capabilities of LLMs. Therefore, we encapsulate optimization tools (such as Bayesian optimization)as standardized function interfaces to quantitatively evaluate the performance improvement of the current algorithmic logic within its hyperparameter space.

**Action and Feedback Loop** Drawing inspiration from optimization iteration patterns in industrial practice, we utilize historical adjustment records as an experiential foundation for subsequent optimization. Each algorithm modification is structurally preserved as a triplet including modification rationale, specific action, and observed results. Before a new round of optimization, these historical experiences are comprehensively presented to the LLM, enabling it to learn from previous attempts, avoid repeating mistakes, and simultaneously identify and replicate successful [43].

**Differential Developer Capability Simulation** To model developers with varying expertise and available resources, we simulate different capability levels by adjusting the computational budget of the agent, rather than by crafting different prompts. We primarily restrict agent capabilities along two dimensions: first, limiting the number of Bayesian optimization calls available for fine-grained parameter tuning; second, constraining the number of reflection iterations for breaking through algorithmic logical boundaries through systematic trial and error. This differential simulation, grounded in resource consumption, enables us to more realistically evaluate the practical value of algorithms under various tuning conditions.

## 3.2 Formalization of Potential

We formalize an algorithm's potential, $\mathcal{P}$, as its performance gain, weighted by a unified environmental distance metric that is derived from the performance profiles of a set of probe algorithms.

**Performance Characteristic Vector.** To quantify an environment's characteristics, we first define an evaluation set of environments $\mathcal{T}$. We then select a small set of representative probe algorithms. For any environment $E_k \in \mathcal{T}$, we run the $n$ probe algorithms to obtain a raw performance score vector $[s_1(E_k), \ldots, s_n(E_k)]$. To eliminate dimensional effects, we normalize each score component across all environments in $\mathcal{T}$:

$$\text{norm}(s_j(E_k)) = \frac{s_j(E_k) - s_{j,\min}}{s_{j,\max} - s_{j,\min}}, \tag{1}$$

where $s_{j,\max} = \max_{E_k \in \mathcal{T}}(s_j(E_k))$ and $s_{j,\min} = \min_{E_k \in \mathcal{T}}(s_j(E_k))$. This process yields an $n$-dimensional normalized performance characteristic vector $V(E_k)$ for each environment $E_k$, which serves as its quantitative fingerprint. The vector is explicitly defined as:

$$V(E_k) = [\text{norm}(s_1(E_k)), \text{norm}(s_2(E_k)), \ldots, \text{norm}(s_n(E_k))]. \tag{2}$$

**Unified Environment Distance and Similarity.** Using these characteristic vectors, the distance between two environments $E_i$ and $E_t$ is defined as the root mean square error (RMSE):

$$\text{dis}(E_i, E_t) = \sqrt{\frac{1}{n} \sum_{j=1}^{n} (V(E_i)_j - V(E_t)_j)^2}. \tag{3}$$

Here, $V(E_i)_j$ denotes the $j$-th component of the vector $V(E_i)$.

The corresponding environment similarity is then defined as:

$$\text{sim}(E_i, E_t) = \max(0, 1 - \text{dis}(E_i, E_t)). \tag{4}$$

**Tuning Potential Definition.** With the environmental similarity metric now formally defined, we can present the complete definition of an algorithm's potential, $\mathcal{P}$. It is the similarity-weighted average performance gain across all test environments. For a given algorithm, let $S_{t,o}$ and $S_{t,c}$ be its original and Crucible-tuned performance in a test environment $E_t$, and let $E_i$ be its ideal environment. The potential is calculated as:

$$\mathcal{P} = \frac{1}{|\mathcal{T}|} \sum_{E_t \in \mathcal{T}} [(S_{t,c} - S_{t,o}) \times \text{sim}(E_i, E_t)]. \tag{5}$$

This formulation ensures that performance gains in environments highly dissimilar to the ideal one are down-weighted, yielding a more robust and fair measure of an algorithm's intrinsic tunability.

### 3.3 Interaction Between `Crucible` and Control Algorithms

The `Crucible` framework supports multiple control tasks, thus designing a standardized interaction interface. In this interface, control algorithms provide two types of information to the LLM: the algorithm code itself and execution logs. The execution logs consist of a series of triplets containing states, actions, and results. Based on this information, the LLM exclusively modifies the control algorithm and obtains new test results by invoking the original execution file. This unified interface effectively standardizes the interaction logic, enabling `Crucible` to flexibly support any type of control algorithm.

The overall interaction logic of the system is as follows: First, the system traverses all preset test environments and executes the LLM optimization cycle in each environment. In this cycle, the system compares the performance of the current algorithm with reference algorithms (which can be overfit learning-based algorithms or theoretically optimal algorithms) and collects cases with significant performance differences. Subsequently, the LLM provides algorithm optimization suggestions based on these difference information. The system implements these suggestions and optionally applies Bayesian optimization methods to further adjust algorithm parameters. When all environments have been traversed, the system enters the evaluation phase. For each algorithm under test, it first determines its ideal environment (e.g., the one yielding the best performance). Then, using the unified performance-characteristic-based metric defined in Section 3.2, it calculates the similarity between each test environment and the ideal environment. Finally, it computes the algorithm's potential by aggregating the similarity-weighted performance gains across all test environments. Appendix B provides a detailed pseudocode of the complete Crucible workflow.

## 4 Evaluation

We begin by outlining our experimental setup in Section 4.1, with complete configuration details deferred to Appendix A. Our evaluation then unfolds in three progressive stages to systematically validate the `Crucible` framework.

First, in Section 4.2, we establish the framework's **effectiveness and generalizability**. We demonstrate its superiority over traditional tuning, validate its performance in a real-world deployment, and confirm its robustness across different LLMs. Building on this, Section 4.3 shows how `Crucible` moves beyond mere performance enhancement to provide **quantitative insights** into the abstract concept of algorithmic potential, revealing the key factors that govern it. Finally, in Section 4.4, we illustrate the **practical impact** of these insights, showing how they guide targeted algorithm design and establish potential as a valuable optimization target.

### 4.1 Experimental Setup

#### 4.1.1 `Crucible`

This research mainly employs API calls to the Claude 3.7 Sonnet [44], with Bayesian optimization serving as a hyperparameter tuning tool. For simulating developers with varying optimization capabilities, we configure the Bayesian iteration count at three distinct levels (0, 10, and 20 iterations) while setting the reflection iteration steps to 1, 2, and 3, respectively. Our evaluation spans a diverse range of control tasks to demonstrate Crucible's generalizability.

#### 4.1.2 Case Studies

We select testbeds from three distinct domains to demonstrate Crucible's generalizability.

**Classic Control.** We use the "CartPole-v1" environment from Gym [45], a standard benchmark in control task. Our evaluation focuses on classic controllers such as Proportional-Integral-Derivative (PID) [46] and Linear-Quadratic Regulator (LQR) [47], comparing their optimized performance against RL-based methods DQN [48].

Table 1: Performance of classic control algorithms on Cart-Pole, tuned by `Crucible`.

| Algorithm | Initial | After 1 Bayes | After 1 LLM | After 2 Bayes | After 2 LLM |
|---|---|---|---|---|---|
| Bang_bang | 34 | 56 | 500 | - | - |
| PID | 34 | 77 | 110 | 271 | 500 |
| LQR | 161 | 500 | - | - | - |
| DQN (Reference) | 500 | - | - | - | - |

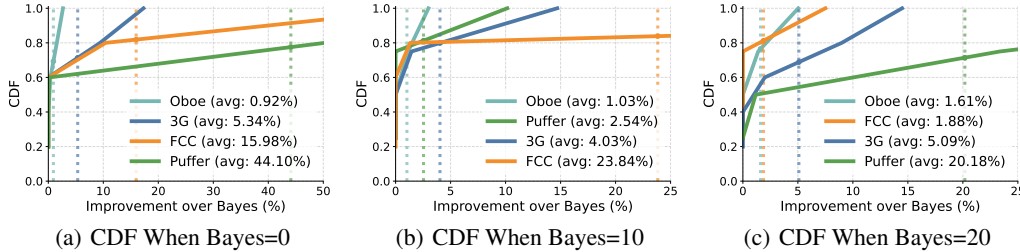

(a) CDF When Bayes=0     (b) CDF When Bayes=10     (c) CDF When Bayes=20

Figure 3: Optimization Space between Crucible and Bayes

**Computer Systems (ABR).** We utilize a widely adopted adaptive bitrate (ABR) simulator [32, 29] with four public network datasets: Oboe [30], FCC [49], 3G [50], and Puffer [51]. The experiments evaluate prominent algorithms, including BBA, MPC, HYB, BOLA, and Pitree, using the Envivio video trace [32] as standardized content. In addition to simulated environments, we conduct validation in a real-world setting using Dash.js [52].

**Computer Systems (Scheduling).** We employ a Spark simulator [53] with TPC-H query tasks [54] to evaluate classical and modern scheduling algorithms, including Shortest Job First (SJF), First-In-First-Out (FIFO), Multi-level feedback (MF), and Tetris [35].

## 4.2 Crucible's Effectiveness and Generalizability

### 4.2.1 Effectiveness: Expanding the Optimization Space

Unlike traditional optimization tools confined to hyperparameter tuning, `Crucible` implements modifications at the algorithmic logic level, expanding the optimization space. An illustration of this is found in the classic Cart-Pole control problem. As shown in Table 1, simple heuristic algorithms like Bang-bang [16] and PID, which initially perform limited, can achieve the optimal score of 500, matching the performance of a complex black-box algorithm like DQN. This improvement, particularly the jump from a score of 56 to 500 for the Bang-bang controller after a single LLM-driven logic modification, is a direct result of altering the core control logic—a performance leap unattainable through mere parameter tuning.

This principle of logic-level enhancement is not an isolated phenomenon but a general advantage that holds true in more complex domains. Across our computer system benchmarks, Figure 3 illustrates the relative improvement ratio achieved by `Crucible`'s enhanced results $S_c$ compared to Bayesian optimization results $S_b$, expressed as $(S_c - S_b)/S_b$. The results demonstrate that `Crucible`'s logic-level improvements consistently yield performance enhancements, achieving gains of up to 44.1% on the Puffer dataset.

However, these logic-level modifications also introduce variability. Our experiments indicate that without the fine-tuning provided by Bayesian optimization, pure LLM-driven adjustments can be unreliable; approximately 60% of test scenarios show no significant performance gains, primarily attributable to the LLM's limitations in fine-grained numerical operations (Figure 3(a)). This highlights the critical synergy within our framework. As Bayesian optimization is incorporated, it not only improves the baseline but also effectively explores the new solution space opened up by the LLM's logic changes. Consequently, the proportion of ineffective test scenarios decreases from 60% to 20% (Figure 3(c)), confirming the powerful combination of logic-level exploration and parameter-level exploitation.

#### 4.2.2 Real World Evaluation

We validate `Crucible` in a real-world ABR scenario using Dash.js over a public WiFi network. We test five heuristic algorithms and the RL-based Pensieve as a reference. The results, presented in Table 2, demonstrate that `Crucible` successfully enhances the performance of heuristic algorithms in this noisy, unpredictable environment. For example, the tuned HYB and BBA algorithms achieve a QoE score of 1.72, outperforming their original versions and even surpassing the RL-based Pensieve (1.66). Conversely, the complex and less interpretable Pitree algorithm exhibits no improvement, reinforcing that `Crucible`'s effectiveness correlates with an algorithm's comprehensibility. These findings provide evidence that the benefits of `Crucible`-guided tuning transfer directly to practical, real-world deployments.

Table 2: QoE of ABR in a real-world Dash.js deployment before and after `Crucible` tuning.

| QoE State | HYB | BBA | BOLA | Pitree | MPC | Pensieve (RL) |
|---|---|---|---|---|---|---|
| Original | 1.40 | 1.56 | 1.20 | 1.73 | 1.72 | 1.66 |
| Crucible-Tuned | 1.72 | 1.72 | 1.54 | 1.73 | 1.79 | - |

#### 4.2.3 Robustness Across Different LLMs

We evaluate the framework using three different models: Claude 3.7 Sonnet (our primary model), the previous generation Claude 3.5 Sonnet, and GPT-4o-mini. As detailed in Table 3, while the absolute final performance varies slightly across models, the overall conclusions remain consistent. All LLMs effectively tune the ABR algorithms, and the relative performance ranking among them remains largely stable. Claude 3.7 Sonnet achieves a higher final score on the HYB algorithm, suggesting that more powerful models can unlock greater potential in certain cases, but other models also deliver significant improvements.

Table 3: Final ABR performance after tuning with different LLMs. All runs were configured with 3 iterations and 20 Bayesian iterations.

| Algorithm | Initial | Claude 3.5 Sonnet | GPT-4o-mini | Claude 3.7 Sonnet |
|---|---|---|---|---|
| BBA | 0.75 | 1.13 | 1.10 | 1.11 |
| BOLA | 1.02 | 1.07 | 1.08 | 1.06 |
| HYB | 0.92 | 1.03 | 1.04 | 1.12 |
| Pitree | 0.31 | 0.35 | 0.37 | 0.36 |
| MPC | 1.07 | 1.09 | 1.10 | 1.09 |

### 4.3 Potential Analysis

In Figure 4, we compare the percentage performance improvements of ABR and scheduling algorithms under different `Crucible` capability settings against their original scores. Regarding optimization tool usage constraints, Figures 4(a) through 4(c) demonstrate a positive correlation between increased Bayesian optimization iterations and significant algorithm performance enhancements. Specifically, when the number of iterations increased from one to two, a notable performance leap occurred—in ABR, the improvement rate with zero Bayesian iterations rose from 9.54% to 29.91%, while in scheduling algorithms, the proportion of scenarios failing to achieve performance improvements decreased from 80% to 60%. Between the two dimensions examined, the ability of optimization tools to unlock algorithmic logic potential has a more pronounced impact on performance enhancement. Furthermore, comparing ABR and scheduling algorithms reveals that due to differences in input state complexity (ABR only involves buffer size and bandwidth, whereas scheduling encompasses complex DAG graph information and node states), LLMs face varying degrees of comprehension challenges, resulting in significantly lower improvement magnitudes for scheduling algorithms compared to ABR algorithms.

We illustrate the detailed potential analysis results with ABR algorithms as an example in Table 4. Through our analysis, we derive two key findings: First, even among algorithms representing simple logic, the HYB algorithm utilizing a broader state space demonstrates significantly greater potential

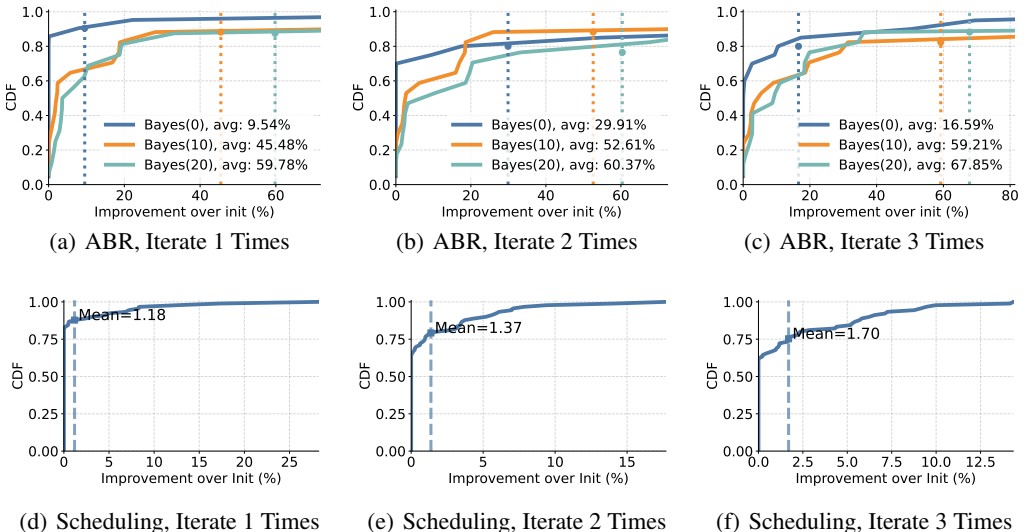

Figure 4: Improvement in Different Ability

than the BBA algorithm using only a single state space, indicating that algorithmic representational capacity substantially influences potential enhancement; Second, as a decision tree algorithm distilled from deep reinforcement learning, Pitree exhibits lower optimization potential despite its initial limited performance, suggesting that the complex logical structure of decision trees may negatively impact potential enhancement. Therefore, researchers aspire to design ABR algorithms that improve comprehensibility, such as ComTree [14].

Table 4: Potential of ABR Algorithms

| Alg | init | 1 Iter | 2 Iter | 3 Iter | Impro | Potential | Ideal |
|------|------|--------|--------|--------|-------|-----------|-------|
| HYB | $0.92 \pm 0.64$ | $0.99 \pm 0.61$ | $1.03 \pm 0.57$ | $1.02 \pm 0.60$ | $0.10 \pm 0.18$ | $0.068 \pm 0.117$ | FCC |
| Pitree | $0.31 \pm 0.10$ | $0.33 \pm 0.10$ | $0.44 \pm 0.27$ | $0.38 \pm 0.09$ | $0.07 \pm 0.12$ | $0.033 \pm 0.022$ | Puffer |
| BOLA | $1.01 \pm 0.45$ | $1.05 \pm 0.46$ | $1.04 \pm 0.45$ | $1.05 \pm 0.45$ | $0.03 \pm 0.06$ | $0.025 \pm 0.032$ | 3G |
| BBA | $0.75 \pm 0.75$ | $1.04 \pm 0.59$ | $0.98 \pm 0.56$ | $1.01 \pm 0.57$ | $0.26 \pm 0.38$ | $0.018 \pm 0.008$ | Oboe |
| MPC | $1.07 \pm 0.52$ | $1.10 \pm 0.59$ | $1.08 \pm 0.53$ | $1.09 \pm 0.54$ | $0.02 \pm 0.04$ | $0.017 \pm 0.014$ | 3G |

## 4.4 From Potential Assessment to Algorithm Optimization

This section explores how to transform `Crucible`'s potential assessments into effective algorithm optimization strategies to enhance ultimate performance.

### 4.4.1 Enhancing Algorithmic Representational Capacity

In the case of ABR algorithms, the results show that the BBA algorithm exhibits both lower optimization potential and inferior final performance compared to the HYB algorithm. This indicates inherent representational limitations in BBA's buffer-only control approach. Based on this insight, we improve BBA to create the BBA_C algorithm (detailed implementation in Appendix C). Specific enhancements include: incorporating current bandwidth as an additional control input and introducing a bandwidth-based control branch to select the highest bitrate that would not cause video stalling.

As shown in Figure 5(a), the enhanced BBA_C's initial performance closely resembles the original BBA algorithm, differing by only 0.5%. From a traditional evaluation perspective, these algorithms appear nearly equivalent. However, after optimization adjustments via `Crucible`, BBA_C's enhanced representational capacity advantage becomes evident. During each optimization iteration, BBA_C consistently outperforms BBA, ultimately achieving a 4% performance improvement. This case demonstrates how enhancing algorithmic representational capacity can increase optimization potential and ultimately improve performance.

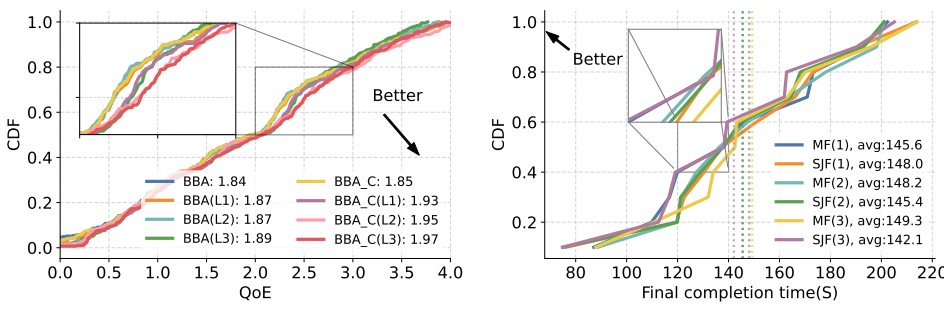

(a) CDF of ABR QoE, Oboe, L1 means 1 iters     (b) CDF of Scheduling FCT, 10*2G input Sizes

Figure 5: CDF After Optimization

### 4.4.2 Simplifying Algorithm Logic

Another important dimension involves improving developers' comprehension capabilities and optimization space by simplifying algorithmic logic. Figure 5(b) presents a typical case in task scheduling: initially, the Multilevel Feedback algorithm outperforms the SJF algorithm; however, after optimization adjustments through `Crucible`, the SJF algorithm achieves shorter task completion times than the supposedly more advanced Multilevel Feedback algorithm.

The fundamental cause of this phenomenon lies in the high-dimensional complexity of input states in DAG scheduling problems. This complexity makes it difficult for LLMs to comprehensively understand the internal logic of complex algorithms, thereby limiting their ability to add appropriate optimization logic to complex algorithms. In contrast, LLMs can more effectively understand and optimize foundational algorithms with simple, clear logic. This finding emphasizes the important value of "simplicity" in algorithm design, particularly in scenarios leveraging AI-assisted optimization.

## 5 Limitations and Broader Impacts

**Limitations** As the first work exploring the potential of control algorithms, this research has two main limitations. First, the stability issue: since we use an LLM as our foundation, different capabilities and versions of LLMs may influence the results. However, we believe this does not diminish the value of assessing algorithmic potential, as different LLMs essentially simulate developers with varying skill levels, making the evaluation, selection, and design of algorithms still practically meaningful. Second, we currently cannot directly modify the internal logic of black-box algorithms; therefore, in this paper, we discuss and analyze decision trees distilled from black-box algorithms. Effectively understanding and adjusting the internal logic of black-box algorithms remains an open challenge, providing direction for future research.

**Broader Impacts** We hope our work can inspire a rethinking of algorithm design, positioning potential as a new optimization direction or even as an optimization metric.

## 6 Conclusion

We introduced `Crucible`, a framework that addresses the gap between algorithm design and practical deployment by quantitatively evaluating Tuning Potential. `Crucible` leverages an LLM agent to simulate developer behavior and introduces a formalized potential metric to quantify this untapped optimization space. Our extensive evaluations—spanning classic control tasks, complex computer systems, and a real-world deployment—demonstrate that `Crucible` not only enhances algorithm performance beyond traditional methods but also establishes tuning potential as a valuable, quantifiable metric. This work advocates for a shift in algorithm design, treating Tuning Potential as a property to be evaluated from the outset, rather than as a post-deployment afterthought. This empowers designers to build adaptable algorithms that maintain long-term value in evolving real-world environments.

**Acknowledgments** We thank the anonymous NeurIPS reviewers for their constructive feedback, which has significantly improved this work exploring new optimization directions. This research was supported by the Beijing National Research Center for Information Science and Technology under Grant BNR2023TD03005-2, the NSERC Discovery Grant, and the Beijing Key Laboratory of Networked Multimedia.

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

| Dataset | Traces | Avg. BW (Mbit/s) | BW Range (Mbit/s) |
|---------|--------|------------------|-------------------|
| 3G | 142 | $1.52 \pm 0.72$ | [0.60, 4.59] |
| Oboe | 428 | $2.77 \pm 1.32$ | [0.34, 5.70] |
| FCC | 264 | $1.33 \pm 0.55$ | [0.19, 3.43] |
| Puffer | 500 | $1.60 \pm 0.88$ | [0.30, 3.60] |

Table 5: Details of Network Trace Datasets

# Appendix

# A    Details Experimental Setup

## A.1    ABR Experimental Setup

### A.1.1    Network Traces

We summarize the characteristics of the four public network traces utilized in this study in Table 5, including the number of traces, average bandwidth per trace, and bandwidth range.

### A.1.2    Video Samples

We use the "EnvivoDash3" video from the "MPEG-DASH reference videos," consistent with previous works [32, 29]. The video is 193 seconds in length and segmented into 4-second chunks with bitrates of {300, 750, 1200, 1850, 2850, 4300} kbps.

### A.1.3    Player Configuration

We employ a classical VoD scenario player simulator as used in [32, 29], with an initial default video quality of 750 kbps.

### A.1.4    Quality of Experience (QoE) Metrics

For the video-on-demand scenario, we adopt the classical $QoE_{lin}$ as our QoE metric [32, 29]. This model is defined as follows:

$$QoE_{lin} = \sum_{n=1}^{N} q(R_n) - \mu \sum_{n=1}^{N} T_n - \sum_{n=1}^{N-1} |q(R_{n+1}) - q(R_n)| \tag{6}$$

where $N$ represents the total number of video segments, $R_n$ denotes the bitrate of the $n$-th segment, and $q(R_n)$ is a function mapping bitrate to perceived user quality. $T_n$ represents the buffering time incurred while downloading the $n$-th segment, and $\mu$ is the weight coefficient for buffering time. The final term penalizes variations in video quality to ensure playback smoothness. Specifically, in our implementation, we set $\mu$ to 4.3.

## A.2    Scheduling Experimental Setup

### A.2.1    Spark Simulator

We utilize a high-fidelity simulator [53] to evaluate the performance of our scheduling policies. This simulator is designed to faithfully emulate the behavior of a real Spark cluster by capturing several key real-world phenomena:

1. **Initial Task Wave Effects**: The first wave of tasks in a particular stage often exhibits slower execution compared to subsequent tasks. This slowdown arises from factors such as Spark's pipelined task execution, just-in-time (JIT) compilation of task code, and initial overheads like establishing TCP connections between executors. The simulator accounts for this by

drawing the runtime of first-wave tasks from a separate distribution distinct from that of later waves.

2. **Executor Startup Delays**: Adding an executor to a Spark job involves starting a new JVM process, which typically incurs a delay of 2–3 seconds. To accurately reflect this behavior, the simulator imposes a startup delay whenever an executor is reallocated across jobs, mirroring the real-world costs associated with executor initialization.

3. **Impact of High Parallelism**: High degrees of parallelism can negatively impact the performance of individual tasks. Wider shuffle operations require more TCP connections and introduce additional computational overhead when merging data from a large number of shards. The simulator captures these effects by sampling task durations from distributions corresponding to different levels of parallelism, provided such data is available.

Through these mechanisms, the simulator effectively replicates the dynamic behavior of a real-world Spark cluster. This enables rigorous testing and validation of scheduling policies under realistic conditions, ensuring that the results are representative of practical deployment scenarios.

### A.2.2  Workload Setup

For the input workload, we use 2GB of data consisting of 10 TPC-H standard query tasks [54]. TPC-H is a benchmark suite widely used to evaluate decision support systems. It consists of complex analytical queries that reflect real-world workloads in distributed data processing environments. Each query involves a combination of computations, such as joins, aggregations, and data filtering, which are represented as Directed Acyclic Graphs (DAGs) of tasks in the Spark environment. The DAG structure introduces dependencies between tasks, making it ideal for testing the efficacy of advanced scheduling strategies.

## B  `Crucible` **Pseudocode**

The detailed interaction logic between `Crucible` and the control algorithm is illustrated in Algorithm 1.

## C  **Optimization for BBA**

We modify BBA to incorporate two control logics based on bandwidth and buffer occupancy. The bandwidth-based control selects the maximum bitrate that does not cause stalling, while the original buffer-based logic selects its own bitrate. The algorithm then chooses the smaller of these two bitrates. The detailed pseudocode can be found in 2.

**Algorithm 1:** `Crucible` Algorithm

**Input:** Algorithm $Alg_{cur}$, Reference algorithm $Alg_{ref}$, Number of LLM adjustments $N_{llm}$, Number of Bayesian optimizations $N_{BO}$, Test environments $Envs$

**Output:** Potentials

**1 Function** `ApplyBayesianOptimization`(*Alg, $N_{BO}$, E*):

  **2**    **if** $N_{BO} > 0$ **then**

  **3**      $params, ranges = LLMIdentifyParameters(Alg)$

  **4**      $Alg, score = BO(Alg, params, ranges, N_{BO}, E)$

  **5**    **else**

  **6**      $score = E(Alg)$

  **7**    **return** $Alg, score$

**8 for** $E^i \in Envs$ **do**

  **9**    $Alg^i_{cur} = Alg_{test}, score^i_{init} = E^i(Alg^i_{cur}), score^i_{cur} = score^i_{init}$

  **10**    $Alg^i_{cur}, score^i_{cur} = $ `ApplyBayesianOptimization`($Alg^i_{cur}, N_{BO}, E^i$)

  **11**    $BadCases^i = \{\}$

  **12**    **for** $j = 1$ **to** $N_{llm}$ **do**

  **13**      /* Compare with reference algorithm and collect bad cases */

  **14**      $newBadCases^i = CompareAlgs(Alg^i_{cur}, Alg_{ref}, E^i)$

  **15**      $BadCases^i = BadCases^i \cup newBadCases^i$

  **16**      /* Get optimization suggestions from LLM */

  **17**      $suggestions^i, reasons^i = LLMOptimizationSuggestions(Alg^i_{cur}, BadCases^i)$

  **18**      /* Apply optimization suggestions */

  **19**      $Alg^i_{new} = ApplySuggestions(Alg^i_{cur}, suggestions^i)$

  **20**      $Alg^i_{new}, score^i_{new} = $ `ApplyBayesianOptimization`($Alg^i_{new}, N_{BO}, E^i$)

  **21**      **if** $score^i_{new} > score^i_{cur}$ **then**

  **22**        /* Score comparison and update */

  **23**        $Alg^i_{cur} = Alg^i_{new}$

  **24**        $score^i_{cur} = score^i_{new}$

**25** /* Find ideal environment and calculate potentials */

**26** $Potentials = \{\}$

**27** $E_{best} = $ null

**28** $Potential_{max} = 0$

**29 for** $E^i \in Envs$ **do**

  **30**    $E_i = GetIdealEnv(Alg_{cur}, Envs)$

  **31**    $D^i = GetDistance(E^i, E_i)$

  **32**    $Potential^i = \frac{score^i_{cur} - score^i_{init}}{D^i}$

  **33**    $Potentials = Potentials \cup \{Potential^i\}$

**34 return** $Potentials$

**Algorithm 2:** BBA_C Algorithm

---

**Input:** Current buffer size $buffer$, chunk length $chunk\_len$, download speed $speed$, video bitrate array $bitrates$, dimension $dim$

**Output:** Selected bitrate index $rate$

1   $cushion \leftarrow 10$ $reservoir \leftarrow$ predefined value $beta \leftarrow 0.95$

2   /* Calculate bandwidth-based bitrate */

3   $bw\_rate \leftarrow 0$ **for** $i = dim - 1, dim - 2, \ldots, 0$ **do**

4      **if** $bitrates[i] \times chunk\_len < beta \times speed \times buffer$ **then**

5         $bw\_rate \leftarrow i$ **break**

6   /* Calculate buffer-based bitrate */

7   $buf\_rate \leftarrow 0$ **if** $buffer < reservoir$ **then**

8      $buf\_rate \leftarrow 0$

9   **else**

10      **if** $buffer \geq reservoir + cushion$ **then**

11         $buf\_rate \leftarrow dim - 1$

12      **else**

13         $buf\_rate \leftarrow \lfloor (dim - 1) \times \frac{buffer - reservoir}{cushion} \rfloor$

14   /* Choose the minimum of both bitrates */

15   $rate \leftarrow \min(bw\_rate, buf\_rate)$

16   $rate \leftarrow \max(0, \min(rate, dim - 1))$

17   **return** $rate$

---

