# OpenReview forum: "Crucible: Quantifying the Potential of Control Algorithms through LLM Agents"
_NeurIPS.cc/2025/Conference — NeurIPS 2025 poster_

### Official Review · Reviewer_V5Ex · 2025-06-21

**Clarity:** 3
**Significance:** 3
**Originality:** 3
**Rating:** 4
**Confidence:** 3

**Summary:**

This paper proposes an evaluation framework called `Crucible`, which is used to quantify the upper limit of the performance that a control algorithm can achieve through expert tuning in actual deployment, i.e., the "potential" of the algorithm. `Crucible` uses large language models (LLMs) as expert agents and systematically evaluates the tunable space of control algorithms in different scenarios through multi-level ability simulations (such as parameter tuning and logical structure optimization) and environment distance normalization methods. The paper selects adaptive bit rate (ABR) control and distributed scheduling as two representative application scenarios, demonstrating that `Crucible` has a wider optimization space compared with traditional parameter tuning methods. It proves the importance of algorithm potential for actual deployment value.

**Questions:**

1. The success rate/failure rate of `Crucible` or the statistics on the effectiveness of the suggestions are not provided. The lack of systematic analysis of the failure rate and error types of the suggestions generated by the LLM makes it difficult to judge whether the successful optimization of `Crucible` is stable and reproducible.
2. All experiments are based on Claude 3.7 Sonnet, and there is no horizontal comparison of the output behavior of `Crucible` on different mainstream LLMs (such as GPT - 4, Gemini).
3. It is recommended that the authors include at least one traditional control scenario (such as trajectory planning, PID control) in future work or extended experiments to verify the generality and depth of the `Crucible` method.

**Ethical Concerns:**

["NO or VERY MINOR ethics concerns only"]

**Final Justification:**

The Crucible framework proposed in this paper combines large language models and Bayesian optimization to quantitatively control the upper limit of algorithm tuning from the perspective of "algorithmic potential", filling the gap in the evaluation of tuning space systems in existing methods. The thesis is reasonably designed and has a clear structure. The multi-level expert capability simulation and environmental distance normalization introduced demonstrate innovation. The author supplemented a wealth of experimental data, including multi-model comparisons, classical control tasks, and real system verifications, enhancing the universality and practical value of the method.

**Limitations:**

Yes.

**Paper Formatting Concerns:**

No.

**Quality:**

3

**Strengths And Weaknesses:**

**1. Strengths**
- **Quality.**
This paper introduces `Crucible`, an innovative framework for evaluating control algorithm tuning potential. The framework uniquely integrates large language models (LLMs) with Bayesian optimization, creating expert agents that simulate developer optimization behaviors in a closed-loop "evaluation-tuning-feedback" system. Through comprehensive experiments with multi-dimensional metrics and reproducible mechanisms, the results demonstrate performance improvements across various environments, empirically validating the framework's effectiveness.
- **Clarity.**
The structure of the paper is reasonable, the discussion is fluent, the objectives are clear, and the logic is complete.
- **Significance.**
The quantitative indicator of "performance improvement space after expert tuning" provided by `Crucible` helps incorporate "tunability" into the algorithm design goals and provides a new evaluation basis for directions such as system optimization, and operator search.
- **Originality.**
In task definition, the "potential" of the control algorithm is taken as a new modeling objective and converted into observable indicators by the LLM expert agent. In method design, a hierarchical expert ability simulation system is constructed, and an environmental distance normalization strategy is introduced. Overall, it demonstrates strong originality and insight.

**2. Weaknesses**
- The absence of failure case analysis for LLM-generated suggestions limits the reader’s ability to evaluate `Crucible` ’s robustness and practical deployability.
- `Crucible` heavily depends on commercial LLM APIs for generating tuning strategies, making it difficult to isolate the contribution of the framework from the behavior of specific models.
- `Crucible` 's key components are built from existing tools, suggesting that its novelty lies more in integration and workflow design than in foundational methodological innovation.

---

> ### Author Rebuttal · Authors · 2025-07-31
>
> Thank you very much for your valuable and constructive feedback. Your comments have helped us examine and refine this work from a deeper perspective. We will respond from three aspects: Crucible's effectiveness rate, horizontal analysis of results from different LLMs, and validation of traditional control scenarios and real-world experiments. We will revise the manuscript and add the newly introduced code to the supplementary materials. If you have any further questions, please feel free to continue the discussion with us.
>
> ## 1. Crucible's Effectiveness Rate
>
> The analysis of Crucible's effectiveness rate can be found in Figures 3 and 4 and the analysis in sections 4.2 and 4.3. We discuss this from two dimensions: 1. The effectiveness rate of control logic adjustments made by LLM 2. The overall effectiveness rate of Crucible (retaining LLM control logic adjustment and Bayesian parameter optimization)
>
> 1. **Effectiveness rate of LLM control logic adjustments.** We compare Crucible with Bayesian optimization results in Figure 3, where Figure 3(a) shows pure LLM control logic adjustment without Bayesian optimization. We can see that the proportion of improvements close to 0 accounts for about 60%, which means the effectiveness rate of pure LLM logic adjustments is only 40%. However, we can see another dimension: when Bayesian optimization becomes available, compared to the same number of Bayesian optimization iterations, the effectiveness rate of LLM control logic adjustments improved to approximately 60%. This indicates that in 20% of scenarios, the general direction of LLM control logic adjustments was correct, but specific parameter selection was problematic. Therefore, with Bayesian optimization, this portion of adjustments also became effective.
>
> 2. **Overall effectiveness rate.** Comparing with initial results in Figure 4, we can see that in ABR scenarios, there are still approximately 20% with improvement rates close to 0. This indicates that in 20% of scenarios, neither logic adjustments nor Bayesian optimization were effective. Scheduling algorithms mainly rely on LLM control logic modifications due to fewer involved parameters. Figure 4(e) shows that because scheduling algorithms often involve high-dimensional inputs, making LLM understanding of inputs more difficult, only about 40% of adjustments are effective.
>
> ## 2. Horizontal Analysis of Results from Different LLMs
>
> In the paper, we used Claude-Sonnet-3.7 interface. LLMs provide us with the opportunity to mimic developers in understanding and adjusting algorithm logic, and we believe this capability is independent of specific LLM models because we primarily leverage the general knowledge of LLMs. Therefore, we did not discuss different LLMs in the manuscript.
>
> We understand the reviewer's concerns about LLM capabilities, so we added Claude-Sonnet-3.5 as a previous generation model and GPT-4o-mini as another mainstream model. We set the iteration count to 3 and Bayesian iterations to 20. The experimental results are as follows (due to rebuttal time constraints, we did not complete the MPC algorithm results):
>
> | Algorithm | Initial | Sonnet-3.5 | GPT-4o-mini | Sonnet-3.7 |
> |-----------|---------|------------|-------------|------------|
> | bb        | 0.7543  | 1.1283     | 1.1024      | 1.1083     |
> | bola      | 1.015   | 1.0661     | 1.0811      | 1.0642     |
> | hyb       | 0.918   | 1.0316     | 1.0377      | 1.1242     |
> | pitree    | 0.3079  | 0.3524     | 0.3671      | 0.3616     |
>
> LLMs do indeed affect the final post-adjustment performance, but the impact magnitude is modest. Claude-Sonnet-3.7's post-adjustment performance is overall superior to Sonnet-3.5 and 4o-mini, averaging 0.02 and 0.018 higher respectively. While we believe algorithm adjustment scenarios could be an interesting new LLM evaluation metric, the focus of this work remains on new optimization and evaluation metrics for control algorithms. Therefore, we believe that analyzing the potential dimension of control algorithms based on any current mainstream model is valuable. An interesting point lies in the HYB algorithm, which is the only one where relative ranking changed. After adjustment by Sonnet-3.5 and 4o-mini, its performance still falls short of BOLA and BBA, but after adjustment by Sonnet-3.7, it achieved optimal performance. This suggests that the logic of the HYB algorithm requires stronger models to understand and adjust.
>
> ## 3. Validation of Traditional Control Scenarios and Real-World Experiments
>
>
> 1. **We have added a classic control task, the cart-pole problem as a representative example.**
>
>      We selected 2 common computer system control tasks that have rich baselines and are widely applied in production systems to demonstrate our practical value. We understand the reviewer's concerns about task generalizability. Therefore, we chose the classic control task of cart-pole problem, so that we cover a range from simple classic tasks to complex real-world systems. We used "CartPole-v1" from Gym as simulated environment and selected bang-bang, PID, and LQR as baseline algorithms while using DQN algorithm as a reference. The experimental results after Crucible adjustment are as follows:
>
>    |           | **Initial** | **After Bayesian** | **After 1st LLM Optimization** | **After Bayesian** | **After 2nd LLM Optimization** | **After Bayesian** |
>    |--|-|-|-|--|-|-|
>    | Bang_bang | 34          | 56                 | 500                             |            -             |                  -                   |          -                |
>    | PID       | 34          | 77                 | 110                             | 271                | 500                            |            -              |
>    | LQR       | 161         | 500                |                    -                  |               -          |                   -                  |          -                |
>    | DQN       | 500         |       -                  |             -                         |           -              |              -                       |                -          |
> The cart-pole problem, as a simple classic task, demonstrates two insights: 1. The importance of potential: simple heuristic algorithms can achieve performance comparable to black-box algorithms (reaching a score of 500); 2. LLM expands the optimization space of traditional adjustment methods by modifying control branches (both PID and bang_bang reached a score of 500 after LLM adjusted their control branches).
>
> 2. **We added real-world ABR experiments based on Dash.**
>
>    To demonstrate our effectiveness in complex tasks of real systems, we conducted real network environment validation based on Dash.js. We deployed Dash.js on the client side, connected to cloud servers via public WiFi, and tested five heuristic algorithms (MPC, BOLA, BBA, Pitree, HYB) as well as the reinforcement learning-based Pensieve as a control. Each algorithm controlled 3 minutes of real playback of Envivio video. Crucible optimized based on real playback logs, configured with 3 iterations and maximum 10 Bayesian iterations. The results are as follows:
>
> | Algorithm    | Original | Crucible-Tuned |
> |--------------|----------|----------------|
> | HYB          | 1.40     | 1.72           |
> | BBA          | 1.56     | 1.72           |
> | BOLA         | 1.20     | 1.54           |
> | Pitree       | 1.73     | 1.73           |
> | MPC          | 1.72     | 1.79           |
> | Pensieve(RL) | 1.66     | -              |
>
>    Real environment results validated two findings: (1) Tuned simple heuristic algorithms (HYB, BBA) can potentially surpass black-box algorithms; (2) Algorithm potential correlates with its expressiveness and interpretability - MPC achieved the highest post-tuning performance, while complex Pitree showed no performance change after tuning.

---

> > ### Author Response · Authors · 2025-08-06
> >
> > Dear Reviewer,
> >
> > Thank you for your valuable feedback. We appreciate it greatly. With only two days left in the rebuttal period, if you have any further comments, suggestions, or questions needing clarification, please feel free to share them. We very appreciate your time and engagement.
> >
> > Best regards,
> >
> > Authors

---

### Official Review · Reviewer_BJf8 · 2025-06-28

**Clarity:** 1
**Significance:** 2
**Originality:** 2
**Rating:** 5
**Confidence:** 4

**Summary:**

This manuscript proposes Crucible, a framework that uses large language models (LLMs) to evaluate the tuning potential of control algorithms, where an algorithm's tuning potential is defined as the performance improvement it could achieve when adjusted by domain experts, relative to the performance of its preset settings. Specifically, Crucible consists of a multi-level expert simulation system whereby an LLM (e.g. Claude 3.7 Sonnet) is prompted with domain knowledge including task description, optimization objectives, and environment overview, and acts as a proxy for human experts to iteratively refine an algorithm’s hyperparameters and logic with variable number of reflective iterations and access to optimization tools. To ensure a fair potential score across different scenarios, an environment distance metric is proposed to normalize performance gains  by how different the test environment is from the preset environment. The proposed framework is applied to two domains from computer systems, namely, adaptive bitrate control and distributed task scheduling, demonstrating that it can uncover a larger performance optimization space than Bayesian optimization alone by suggesting additions and removal of control logic. The analysis indicates that there are two key factors influencing an algorithm’s tunability: its representational capacity and comprehensibility.

**Questions:**

1. What exactly were the prompts and LLM outputs that drove Crucible’s algorithm improvements? The paper should include concrete examples of the LLM’s input and output (e.g. a sample prompt and the suggestions it gave for improving an algorithm).

2. How sensitive are Crucible’s results to the choice of LLM (e.g., Claude vs. GPT-4)? Would different LLMs yield significantly different tuning outcomes? It would be beneficial to include a sensitivity analysis across multiple LLMs to assess robustness.

3. The Crucible agent simulates different expert skill levels by limiting tool iterations and reflection steps, but how do we know these settings realistically correspond to expert abilities? Can the authors justify or validate this design?

4. The manuscript uses Wasserstein distance for ABR and a proportion-based metric for scheduling. Can you include the mathematical equations for these metrics? How these metrics were chosen, and how sensitive are the results to these choices?

5. How well do the LLMs simulate real expert behavior? Are there any benchmarks or validations comparing LLM-generated tuning strategies with those from human experts? Have the authors considered comparing against real expert adjustments or other algorithm-tuning techniques? For example, for one of the simpler scenarios, one could ask a human expert to manually fine-tune the algorithm and see how much improvement they get, to check if Crucible is on par or discovering similar enhancements.

**Ethical Concerns:**

["NO or VERY MINOR ethics concerns only"]

**Final Justification:**

I have carefully reviewed the author response and all subsequent discussions. The authors have addressed the main concerns raised in my initial reviews, including prompt specification, sensitivity to LLM choice, and distance metrics used. Therefore, I am increasing my overall score to reflect the improved clarity and completeness of the submission.

**Limitations:**

yes

**Quality:**

2

**Strengths And Weaknesses:**

* Strengths

1. The manuscript addresses a practical yet under-explored issue: algorithms in production are often tuned by experts, but research usually ignores this tunability aspect. Through two case studies from computer systems, it is shown that tuning potential is crucial for real-world performance and that a well-tuned simple algorithm can rival or exceed more sophisticated ones.

2. The Crucible framework is innovative in its use of an LLM as a proxy for human developers, which provides a way to simulate expert tuning and human-in-the-loop evaluations at scale without the cost of human trials.

3. It is interesting to systematically show that combining LLM-based logic updates with automatic hyperparameter tuning using bayesian optimization yields the best results.

* Weaknesses

1. The manuscript suffers from a lack of clarity and detail in its presentation. Specifically, the Crucible involves complex interactions (prompts, iterative loops), but the description in the paper is often high-level and omits key specifics. For example, the framework relies on system prompts to inject domain knowledge into the LLM, yet the actual prompt contents and examples of advice given by the LLM are neither provided nor discussed. Similarly, the environment distance metrics are not provided, motivated, or well-explained. While the algorithm pseudo-code for Crucible (Algorithm 1) is included, it is not discussed in depth beyond a formal outline, which makes it harder to appreciate the technicalities of the design.

2. The manuscript doesn’t discuss how results might vary with a different or weaker LLM, which is important given that it's the core technology in the proposed framework. Furthermore, it is not clear how well Crucible generalizes to other domains like robotics or financial systems.

3. The manuscript lacks an ablation study to assess how different design decision in the crucible framework impact the overall performance.

4. The significance of the potential metric would be stronger if validated against human expert tuning in practice. Currently, it’s an interesting proxy, but we don’t know if algorithms that Crucible deems high potential indeed are really easier for real engineers to tune.

---

> ### Author Rebuttal · Authors · 2025-07-31
>
> Thank you very much for your valuable and constructive feedback. Your comments have helped us examine and refine this work from a deeper perspective. We will respond from five aspects: Crucible's prompt and interaction details, analysis of results from different LLMs, ablation study explanations, unified environment distance metrics, and comparison between Crucible and human experts. We will revise the manuscript and add the newly introduced code to the supplementary materials. If you have any further questions, please feel free to continue the discussion with us.
>
> ## 1. Exact Prompts and LLM Outputs Driving Crucible's Algorithm Improvements
>
> We understand concerns about clarity, but since our article emphasizes LLMs providing opportunities to examine new optimization directions, we focused on high-level concepts rather than specific prompts. Our multi-level capability aspects use workflow definitions like tool call frequency and LLM iteration counts. Prompts mainly introduce basic task knowledge and interaction interfaces. We believe our work should be decoupled from specific prompt designs - readers can easily design equivalent or better prompts.
>
> Below we provide the general prompt structure using ABR algorithms as an example. The specific implementation details can be found in the supplementary materials (Crucible/abr/server.py for ABR tasks and Crucible/scheduling/server.py for scheduling tasks).
>
> **Tool calling prompt structure:**
>
> ```python
> Please analyze the following ABR algorithm code and identify parameters for Bayesian optimization. Return results in JSON format with parameter name, type, range, current value, and description.
>
> ```
> **Background and optimization prompt structure:**
> ```python
> I need you to optimize a bitrate adaptive algorithm with 6 bitrate levels [300,750,1200,1850,2850,4300]kbps.  Current code:{code}. Current reward: {current_reward}. Optimize history is {history}. Please analyze the code and suggest improvements. First explain improvement ideas, then provide complete modified abr function code.
>
> ```
> ## 2. Analysis of Results from Different LLMs
>
> In the paper, we used Claude-Sonnet-3.7 interface. LLMs provide us with the opportunity to mimic developers in understanding and adjusting algorithm logic, and we believe this capability is independent of specific LLM models because we primarily leverage the general knowledge of LLMs.
>
> We understand the reviewer's concerns about LLM capabilities, so we added Claude-Sonnet-3.5 as a previous generation model and GPT-4o-mini as another mainstream model. We set the iteration count to 3 and Bayesian iterations to 20. The experimental results are as follows (due to rebuttal time constraints, we did not complete the MPC algorithm results):
>
> | Alg| Initial | Sonnet-3.5 | GPT-4o-mini | Sonnet-3.7 |
> |-|-|-|-|-|
> | bb        | 0.7543  | 1.1283     | 1.1024      | 1.1083     |
> | bola      | 1.015   | 1.0661     | 1.0811      | 1.0642     |
> | hyb       | 0.918   | 1.0316     | 1.0377      | 1.1242     |
> | pitree    | 0.3079  | 0.3524     | 0.3671      | 0.3616     |
>
> LLMs do indeed affect the final post-adjustment performance, but the impact magnitude is modest. Sonnet-3.7's post-adjustment performance is overall superior to Sonnet-3.5 and 4o-mini, averaging 0.02 and 0.018 higher respectively. While we believe algorithm adjustment scenarios could be an interesting new LLM evaluation metric, the focus of this work remains on new optimization and evaluation metrics for control algorithms. Therefore, we believe that analyzing the potential dimension of control algorithms based on any current mainstream model is valuable. An interesting point lies in the HYB algorithm, which is the only one where the relative ranking changed. After adjustment by Sonnet-3.5 and 4o-mini, its performance still falls short of BOLA and BBA, but after adjustment by Sonnet-3.7, it achieved optimal performance. This suggests that the logic of the HYB algorithm requires stronger models to understand and adjust.
>
> ## 3. Ablation Study
>
> Crucible improves algorithms through LLM understanding of algorithm modifications to control loops and calling Bayesian optimization methods. We compared improvements over Bayesian optimization in section 4.2 to show that Crucible transcends the limitations of traditional parameter optimization and achieves higher post-adjustment results through logic adjustments. Additionally, Figure 3 compares the different benefits brought by using and not using Bayesian methods to demonstrate the value of tool calling. Comparing Figure 3(c) with Figure 3(a), when Bayesian optimization becomes available, the effectiveness rate of control logic adjustments made by LLM improved from approximately 40% to 60% compared to the same number of Bayesian optimization iterations. This indicates that in about 20% of scenarios, the general direction of control logic adjustments made by LLM was correct, but the specific parameter selection was problematic. Therefore, with Bayesian optimization, this portion of adjustments also became effective.
>
> ## 4. Unified Environment Distance Metrics
>
> We extended the score-based ratio method used as environmental distance for scheduling algorithms to form a unified performance characteristic-based distance measurement approach to improve the applicability of our work.
>
> ### **Theoretical Derivation and Formal Definition**
>
> **Essential Characterization of Environment**
>
> For control algorithms, the essence of an "environment" is not its physical parameters (such as bandwidth or queue length), but the **performance impact patterns** that the environment produces on different algorithms. Based on this insight, we propose a unified environment distance measurement method.
>
> **Step 1: Constructing Performance Characteristic Vector V(E)**
>
> To capture the performance impact pattern of environment E, we define an evaluation environment set $T$, including the ideal environment $E_i$ and a collection of representative test environments $\{E_t\}$. We select $n$ representative "probe algorithms" following principles of diversity and classicality:
>
> - **ABR tasks**: Select model-based MPC and heuristic-based BBA
> - **Scheduling tasks**: Select classic FIFO and SJF
>
> For environment $E_k \in T$, we run probe algorithms to obtain raw performance scores $s_1(E_k), s_2(E_k), ..., s_n(E_k)$. To eliminate dimensional effects, we perform normalization:
>
> $$s_{j,max} = \max_{E_k \in T}(s_j(E_k)), \quad s_{j,min} = \min_{E_k \in T}(s_j(E_k))$$
>
> $$\text{norm}(s_j(E_k)) = \frac{s_j(E_k) - s_{j,min}}{s_{j,max} - s_{j,min}}$$
>
> Thus, environment $E_k$ is represented as an $n$-dimensional normalized **performance characteristic vector** $V(E_k)$.
>
> **Step 2: Defining Unified Environment Distance**
>
> The distance between two environments $E_i$ and $E_t$ uses root mean square error:
>
> $$\text{dis}(E_i, E_t) = \sqrt{\frac{1}{n} \sum_{j=1}^{n} (V(E_i)_j - V(E_t)_j)^2}$$
>
> The corresponding environment similarity is:
> $$\text{sim}(E_i, E_t) = 1 - \text{dis}(E_i, E_t)$$
>
> **Improved Tuning Potential Definition**
>
> Algorithm potential $P$ is defined as similarity-weighted average performance gain:
>
> $$P = \frac{1}{|T|} \sum_{t \in T} [(S_{tc} - S_{to}) \times \text{sim}(E_i, E_t)]$$
>
> where $S_{tc}$ and $S_{to}$ are the post-tuning and pre-tuning performance, respectively.
>
> ### **Recalculated Results**
>
> We recalculated Table 1 in the manuscript based on the unified metric (sorted by "Performance+Potential"):
>
> | Alg| Potential | Performance+Potential |
> |-|-|-|
> | MPC       | 0.0170    | 1.0862              |
> | BOLA      | 0.0250    | 1.0400              |
> | HYB       | 0.0679    | 0.9859              |
> | BBA       | 0.0181    | 0.7724              |
> | Pitree    | 0.0329    | 0.3407              |
>
> We can still observe that: 1. The HYB algorithm based on both buffer and bandwidth has higher potential than BOLA and BBA which are solely buffer-based; 2. The HYB algorithm with simple single-step logic has higher potential than the multi-step decision-making MPC.
>
> ## 5. Comparison Between Crucible and Human Experts
> Thanks for your suggestion regarding real experts, but if real experts were introduced, how to eliminate the subjective uncertainty of the real experts themselves would become a new problem. Therefore, this work demonstrates the role of Crucible through two parts: first, by comparing with optimization tools and existing algorithms that have more complete algorithmic logic to show Crucible's adjustment ability; and second, by showing how the insights brought by Crucible can be used to redesign algorithms.
>
> Figure 1(b) shows that we compared four algorithms: HYB, HYB(B), HYB(C), and MPC. HYB(B) is the result after Bayesian optimization. Both MPC and HYB algorithms use buffer size and bandwidth for decision-making, but MPC uses multi-step model prediction while HYB uses single-step prediction. Therefore, we can view HYB(B) and MPC as human experts' tuning strategies through parameter optimization and control logic modification respectively. The final performance is HYB(C)>MPC>HYB(B)>HYB. This demonstrates two things: 1. Algorithm adjustment is not merely parameter adjustment - HYB after Bayesian optimization still falls short of logic-adjusted MPC; 2. Crucible-adjusted HYB(C) achieved the highest performance through both control logic and parameter adjustments, demonstrating the importance of exploring algorithm potential and the comprehensiveness of Crucible's tuning strategy.
>
> Furthermore, we discussed in section 4.4 how to use the knowledge analyzed by LLM in the potential quantification process to guide algorithm optimization. For example, by expanding algorithm expression space (BBA_C) and simplifying algorithm control logic (from MF to SJF). This part represents the discussion of combining Crucible's potential quantification capability with human expert knowledge, where we obtained insights on how to design algorithms with higher potential from Crucible's adjustment process.

---

> > ### Comment · Reviewer_BJf8 · 2025-08-04
> >
> > Thank you for your response. I appreciate the clarifications provided in the rebuttal, and I find that most of my concerns have been addressed satisfactorily. I encourage the authors to incorporate these clarifications into the revised manuscript for improved clarity and completeness.

---

> > > ### Author Response · Authors · 2025-08-05
> > >
> > > Thank you very much for your review. Your suggestions regarding comparisons across different LLMs, mathematical formulations for environmental distance, and comparisons with human experts have been extremely helpful in improving this work that explores new optimization directions in traditional domains, and we will incorporate these into our revised manuscript. We welcome any further questions or discussions you may have, as they would greatly help us improve this manuscript.

---

### Official Review · Reviewer_Cp8B · 2025-07-01

**Clarity:** 3
**Significance:** 3
**Originality:** 3
**Rating:** 4
**Confidence:** 3

**Summary:**

The paper proposes the Crucible framework, which is the first to use LLM to simulate multi-level experts to tune the control algorithm, thereby calculating the tuning potential of the control algorithm. Through tests in two classic fields, ABR and scheduling algorithms, the effectiveness of the framework was verified, and the importance of the algorithm's tunability as a new optimization goal was explored, which can be customized according to real application scenarios.

**Questions:**

1. The article mentioned that different LLMs will affect the final results. Can you add corresponding comparative tests? I think this is very helpful for the effectiveness of the algorithm.
2. The scheduling algorithm uses a score-based ratio as the environmental distance, which is tricky. Can more analysis or more detailed environmental distance modeling be added?

**Ethical Concerns:**

["NO or VERY MINOR ethics concerns only"]

**Final Justification:**

I sincerely apologize for being offline for the past two weeks due to work commitments. Prior to going offline, I had thoroughly reviewed the authors' rebuttal. At that time, my concerns were adequately addressed, and I deemed no additional experiments necessary to prove generalization or complex control algorithms. Therefore, I submitted an ​​expedited review response​​ directly through the button.

Tonight, upon carefully examining the authors' replies to other reviewers, I observed ​​consistent concerns​​ across the reviews. Notably, the supplementary experiments added by the authors in response to different reviewers appear ​​largely the same​​.

Nonetheless, given that my primary concerns have been resolved, I am willing to revise my score from ​​Borderline Reject​​ to ​​Borderline Accept​​.

**Limitations:**

yes

**Quality:**

3

**Strengths And Weaknesses:**

Pros:
1. Using "tunability" as a new dimension for control algorithm design and evaluation hits the pain points in the industrial deployment process.
2. Using LLM as an expert agent to simulate actual developers to adjust the algorithm at the logic and parameter levels is novel and practical.
3. The detailed pseudo-code of Crucible is provided in the appendix.

Cons:
1. Currently, Crucible does not support optimization of the internal logic of black-box algorithms (it can only adjust the distillation version of the decision tree), which is a common situation for many industrial algorithms in reality.
2. This article mainly focuses on two tasks: ABR and scheduling. The experimental results are not comprehensive enough.

---

> ### Author Rebuttal · Authors · 2025-07-31
>
> Thank you very much for your valuable and constructive feedback. Your comments have helped us examine and refine this work from a deeper perspective. Regarding the optimization of internal logic for black-box algorithms, this can indeed be challenging for production systems because their black-box nature makes their logic incomprehensible to developers. As the first work to explore the new dimension of control algorithm potential, we have chosen to focus primarily on heuristic algorithms, and for black-box algorithms, we opt to distill them into decision trees for analysis. Our response mainly addresses three aspects: addition of classic control scenarios and real-world system validation, unified detailed environment distance modeling analysis, and horizontal comparison of mainstream LLMs with different capabilities. We will revise the manuscript and add the newly introduced code to the supplementary materials. If you have any further questions, please feel free to continue the discussion with us.
>
> ## **1. Regarding the Focus on ABR and Scheduling with Insufficient Comprehensive Experimental Results**
>
> 1. **We have added a classic control task, the cart-pole problem as a representative example.**
>
>      We selected 2 common computer system control tasks that have rich baselines and are widely applied in production systems to demonstrate our practical value. We understand the reviewer's concerns about task generalizability. Therefore, we chose the classic control task of cart-pole problem, so that we cover a range from simple classic tasks to complex real-world systems. We used "CartPole-v1" from Gym as simulated environment and selected bang-bang, PID, and LQR as baseline algorithms while using DQN algorithm as a reference. The experimental results after Crucible adjustment are as follows:
>
>    |           | **Initial** | **After Bayesian** | **After 1st LLM Optimization** | **After Bayesian** | **After 2nd LLM Optimization** | **After Bayesian** |
>    |--|-|-|-|--|-|-|
>    | Bang_bang | 34          | 56                 | 500                             |            -             |                  -                   |          -                |
>    | PID       | 34          | 77                 | 110                             | 271                | 500                            |            -              |
>    | LQR       | 161         | 500                |                    -                  |               -          |                   -                  |          -                |
>    | DQN       | 500         |       -                  |             -                         |           -              |              -                       |        -          |
> The cart-pole problem, as a simple classic task, demonstrates two insights: 1. The importance of potential: simple heuristic algorithms can achieve performance comparable to black-box algorithms (reaching a score of 500); 2. LLM expands the optimization space of traditional adjustment methods by modifying control branches (both PID and bang_bang reached a score of 500 after LLM adjusted their control branches).
>
> 2. **We have added real-world ABR experiments based on Dash.**
>
>    To demonstrate our effectiveness in complex tasks of real systems, we conducted real network environment validation using Dash.js. We deployed Dash.js on the client side, connected to cloud servers via public WiFi, and tested five heuristic algorithms (MPC, BOLA, BBA, Pitree, HYB) as well as the reinforcement learning-based Pensieve as a control. Each algorithm controlled 3 minutes of real playback of Envivio video. Crucible performed optimization based on real playback logs, configured for 3 iterations and a maximum of 10 Bayesian iterations. The results are shown in the following table:
>
>    | Algorithm    | Original | Crucible-Tuned |
>    |-|-|-|
>    | HYB          | 1.40     | 1.72           |
>    | BBA          | 1.56     | 1.72           |
>    | BOLA         | 1.20     | 1.54           |
>    | Pitree       | 1.73     | 1.73           |
>    | MPC          | 1.72     | 1.79           |
>    | Pensieve(RL) | 1.66     | -              |
>
>    The real-world results validate two findings: (1) tuned simple heuristic algorithms (HYB, BBA) can outperform black-box algorithms; (2) algorithm potential correlates with its expressiveness and interpretability, with MPC achieving the highest post-tuning performance while complex Pitree showed no performance change after tuning.
>
> ## **2. Unified Detailed Environment Distance Modeling Analysis**
>
> Thank you for appreciating the environment distance method for scheduling algorithms. This inspired us to extend the score-based ratio method used as environmental distance for scheduling algorithms to form a unified performance characteristic-based distance measurement approach.
>
> ### **Theoretical Derivation and Formal Definition**
>
> **Essential Characterization of Environment**
>
> For control algorithms, the essence of an "environment" is not its physical parameters (such as bandwidth or queue length), but the **performance impact patterns** that the environment produces on different algorithms. Based on this insight, we propose a unified environment distance measurement method.
>
> **Step 1: Constructing Performance Characteristic Vector V(E)**
>
> To capture the performance impact pattern of environment E, we define an evaluation environment set $T$, including the ideal environment $E_i$ and a collection of representative test environments $\{E_t\}$. We select $n$ representative "probe algorithms" following principles of diversity and classicality:
>
> - **ABR tasks**: Select model-based MPC and heuristic-based BBA
> - **Scheduling tasks**: Select classic FIFO and SJF
>
> For environment $E_k \in T$, we run probe algorithms to obtain raw performance scores $s_1(E_k), s_2(E_k), ..., s_n(E_k)$. To eliminate dimensional effects, we perform normalization:
>
> $$s_{j,max} = \max_{E_k \in T}(s_j(E_k)), \quad s_{j,min} = \min_{E_k \in T}(s_j(E_k))$$
>
> $$\text{norm}(s_j(E_k)) = \frac{s_j(E_k) - s_{j,min}}{s_{j,max} - s_{j,min}}$$
>
> Thus, environment $E_k$ is represented as an $n$-dimensional normalized **performance characteristic vector** $V(E_k)$.
>
> **Step 2: Defining Unified Environment Distance**
>
> The distance between two environments $E_i$ and $E_t$ uses root mean square error:
>
> $$\text{dis}(E_i, E_t) = \sqrt{\frac{1}{n} \sum_{j=1}^{n} (V(E_i)_j - V(E_t)_j)^2}$$
>
> The corresponding environment similarity is:
> $$\text{sim}(E_i, E_t) = 1 - \text{dis}(E_i, E_t)$$
>
> **Improved Tuning Potential Definition**
>
> Algorithm potential $P$ is defined as similarity-weighted average performance gain:
>
> $$P = \frac{1}{|T|} \sum_{t \in T} [(S_{tc} - S_{to}) \times \text{sim}(E_i, E_t)]$$
>
> where $S_{tc}$ and $S_{to}$ are the post-tuning and pre-tuning performance, respectively.
>
> ### **Recalculated Results**
>
> Results for Table 1 in the manuscript recalculated based on the unified metric (sorted by "Performance+Potential"):
>
> | Alg | Potential | Performance+Potential |
> |-|-|-|
> | MPC       | 0.0170    | 1.0862              |
> | BOLA      | 0.0250    | 1.0400              |
> | HYB       | 0.0679    | 0.9859              |
> | BBA       | 0.0181    | 0.7724              |
> | Pitree    | 0.0329    | 0.3407              |
>
> We can still observe that: 1. The HYB algorithm based on both buffer and bandwidth has higher potential than BOLA and BBA which are solely buffer-based; 2. The HYB algorithm with simple single-step logic has higher potential than the multi-step decision-making MPC.
>
> ## **3. Horizontal Comparison of Mainstream LLMs with Different Capabilities**
>
> In the paper, we used the Claude-Sonnet-3.7 interface. LLMs provide us with the opportunity to mimic developers in understanding and adjusting algorithm logic, and we believe this capability is independent of specific LLM models because we primarily leverage the general knowledge of LLMs. Therefore, we did not discuss different LLMs in the manuscript.
>
> We understand the reviewer's concerns about LLM capabilities, so we added Claude-Sonnet-3.5 as a previous generation model and GPT-4o-mini as another mainstream model. We set the iteration count to 3 and Bayesian iterations to 20. The experimental results are as follows (due to rebuttal time constraints, we did not complete the MPC algorithm results):
>
> | Alg | Initial Performance | Claude-Sonnet-3.5 Final Performance | GPT-4o-mini Final Performance | Claude-Sonnet-3.7 Final Performance |
> |-|-|-|-|-|
> | bb        | 0.7543              | 1.1283                               | 1.1024                         | 1.1083                               |
> | bola      | 1.015               | 1.0661                               | 1.0811                         | 1.0642                               |
> | hyb       | 0.918               | 1.0316                               | 1.0377                         | 1.1242                               |
> | pitree    | 0.3079              | 0.3524                               | 0.3671                         | 0.3616                               |
>
> LLMs do indeed affect the final post-adjustment performance, but the impact magnitude is modest. Claude-Sonnet-3.7's post-adjustment performance is overall superior to Sonnet-3.5 and 4o-mini, averaging 0.02 and 0.018 higher respectively. While we believe algorithm adjustment scenarios could be an interesting new LLM evaluation metric, the focus of this work remains on new optimization and evaluation metrics for control algorithms. Therefore, we believe that analyzing the potential dimension of control algorithms based on any current mainstream model is valuable. An interesting point lies in the HYB algorithm, which is the only one where relative ranking changed. After adjustment by Sonnet-3.5 and 4o-mini, its performance still falls short of BOLA and BBA, but after adjustment by Sonnet-3.7, it achieved optimal performance. This suggests that the logic of the HYB algorithm requires stronger models to understand and adjust.

---

### Official Review · Reviewer_4H7x · 2025-07-03

**Clarity:** 3
**Significance:** 3
**Originality:** 3
**Rating:** 4
**Confidence:** 3

**Summary:**

The paper introduces the Crucible framework, a novel approach to quantify the "tuning potential" of control algorithms, defined as the performance improvement achievable through parameter and logic adjustments by domain experts in production environments. Crucible leverages large language models to simulate developers of varying expertise levels, using multi-level expert simulation and environment distance metrics (Wasserstein distance for ABR, initial performance ratio for scheduling). The framework evaluates adaptive bitrate (ABR) algorithms and distributed scheduling algorithms on standard datasets, using domain-specific metrics (QoE for ABR, flow completion time (FCT) for scheduling). Results demonstrate Crucible’s ability to quantify tuning potential, revealing the impact of representation capacity and logic complexity on optimization. The paper discusses LLM stability, limitations in improving black-box algorithms, and societal impacts.

**Questions:**

1. The tuning potential metric (Section 3.2) uses task-specific environment distance metrics (Wasserstein for ABR, ratio-based for scheduling), lacking a unified mathematical definition. Can you provide a formal derivation or a generalized metric that ensures cross-task comparability? For example, could a single distance metric (e.g., based on performance distributions) be applied across domains?
2. The multi-level expert simulation uses Bayesian iterations and reflection steps to model varying expertise levels, but the basis for capability delineation is unclear. Can you specify the quantitative criteria for these levels and validate their impact on results (e.g., through ablation studies)?
3. Experiments rely on simulated environments without real-world production system validation. Can you provide results from real-world or near-real-world settings? How do these results compare to simulations?
4. Comparisons with existing optimization methods (e.g., hyperparameter tuning, reinforcement learning) are limited, weakening originality claims. Can you provide a detailed comparison with methods like Bayesian optimization or RL-based tuning?

**Ethical Concerns:**

["NO or VERY MINOR ethics concerns only"]

**Final Justification:**

The authors' rebuttal provides a thoughtful and well-organized response that clarifies several key aspects of the paper. The effort to unify the environment distance metric through performance-based characterization is a meaningful step toward improving theoretical soundness, though a more formal treatment would be beneficial in the future. The inclusion of preliminary real-world validation using Dash.js adds practical value and supports the applicability of Crucible beyond simulations. The explanation of the multi-level expert simulation mechanism—grounded in computational resource usage rather than prompt design—clarifies the modeling choices and aligns with the workflow. Additionally, the expanded comparison with traditional optimization and RL-based methods helps better situate Crucible within the broader literature. While some concerns remain partially open, particularly regarding formalization and generalizability, the rebuttal has strengthened the overall presentation and reinforced the contribution's relevance. I maintain my original positive score.

**Limitations:**

The paper demonstrates Crucible’s applicability to ABR and scheduling but does not discuss the limitations of extending the framework to other domains (e.g., robotics, real-time systems) where control algorithms may have different constraints or data distributions. The task-specific environment distance metrics (Wasserstein for ABR, ratio-based for scheduling) may not generalize well.

**Quality:**

3

**Strengths And Weaknesses:**

**Strengths:**

- This paper introduces the concept of “tuning potential” for the first time and quantifies it through expert simulation based on large language models (LLMs), environmental distance regularization (ABR uses Wasserstein distance, scheduling uses ratio benchmarks), and a standardized interaction interface (Section 3.3), providing a pioneering method for the interdisciplinary field of control algorithms and machine learning.
- The application of Crucible in ABR and scheduling demonstrates cross-domain generalizability. Results (e.g., Figure 5(b), showing that optimized SJF outperforms multi-level feedback) provide new insights for algorithm design, emphasizing the importance of logical simplification and representational capability.
- The study encompasses multiple algorithms (ABR: BBA, MPC, HYB, BOLA, Prince; scheduling: SJF, FIFO, Round Robin, Tetris) and standard datasets (Obee, FCC, Puffer, Spark), using domain-specific metrics (QoE, FCT) and validated through Bayesian optimization (0, 10, 20 iterations). Figures 3 and 5 validate the framework's effectiveness.

**Weaknesses:**

- The definition of optimization potential (Section 3.2) relies on environment distance normalization, and the definition of environment distance varies depending on the task, lacking a unified mathematical definition, which limits its theoretical persuasiveness.
- Multi-level expert simulation employs Bayesian iteration (0, 10, 20) and reflection steps (1, 2, 3), but the basis for defining capabilities (e.g., prompt design) is not explicitly stated, potentially affecting reproducibility.
- Experiments rely on simulated environments (ABR simulator, Spark simulator) without validation on real-world production systems, limiting their credibility.
- Comparisons with existing optimization methods (e.g., hyperparameter tuning, reinforcement learning) are incomplete, weakening the claim of originality.

---

> ### Author Rebuttal · Authors · 2025-07-30
>
> Thank you very much for your valuable and constructive feedback. Your comments have helped us examine and refine this work from a deeper perspective. Our response addresses four main aspects: unified environment distance definition, real-world environment validation, multi-level expert simulation mechanism explanation, and comparison with existing optimization methods. We will revise the manuscript and add the newly introduced code to the supplementary materials. If you have any further questions, please feel free to continue the discussion with us.
>
> ## **1. Unified Environment Distance Metric**
>
> Your observation regarding the lack of a unified mathematical definition for environment distance is very insightful and prompted us to re-examine the theoretical foundation of "tuning potential." We believe that a unified, cross-task metric would significantly enhance both the theoretical value and practical applicability of our framework. Therefore, we have extended the score-based ratio method used as environmental distance for scheduling algorithms to form a unified performance characteristic-based distance measurement approach.
>
>
> ### **Theoretical Derivation and Formal Definition**
>
> **Essential Characterization of Environment**
>
> For control algorithms, the essence of an "environment" is not its physical parameters (such as bandwidth or queue length), but the **performance impact patterns** that the environment produces on different algorithms. Based on this insight, we propose a unified environment distance measurement method.
>
> **Step 1: Constructing Performance Characteristic Vector V(E)**
>
> To capture the performance impact pattern of environment E, we define an evaluation environment set $T$, including the ideal environment $E_i$ and a collection of representative test environments $\{E_t\}$. We select $n$ representative "probe algorithms" following principles of diversity and classicality:
>
> - **ABR tasks**: Select model-based MPC and heuristic-based BBA
> - **Scheduling tasks**: Select classic FIFO and SJF
>
> For environment $E_k \in T$, we run probe algorithms to obtain raw performance scores $s_1(E_k), s_2(E_k), ..., s_n(E_k)$. To eliminate dimensional effects, we perform normalization:
>
> $$s_{j,max} = \max_{E_k \in T}(s_j(E_k)), \quad s_{j,min} = \min_{E_k \in T}(s_j(E_k))$$
>
> $$\text{norm}(s_j(E_k)) = \frac{s_j(E_k) - s_{j,min}}{s_{j,max} - s_{j,min}}$$
>
> Thus, environment $E_k$ is represented as an $n$-dimensional normalized **performance characteristic vector** $V(E_k)$.
>
> **Step 2: Defining Unified Environment Distance**
>
> The distance between two environments $E_i$ and $E_t$ uses root mean square error:
>
> $$\text{dis}(E_i, E_t) = \sqrt{\frac{1}{n} \sum_{j=1}^{n} (V(E_i)_j - V(E_t)_j)^2}$$
>
> The corresponding environment similarity is:
> $$\text{sim}(E_i, E_t) = 1 - \text{dis}(E_i, E_t)$$
>
> **Improved Tuning Potential Definition**
>
> Algorithm potential $P$ is defined as similarity-weighted average performance gain:
>
> $$P = \frac{1}{|T|} \sum_{t \in T} [(S_{tc} - S_{to}) \times \text{sim}(E_i, E_t)]$$
>
> where $S_{tc}$ and $S_{to}$ are the post-tuning and pre-tuning performance, respectively.
>
> ### **Recalculated Results**
>
> Results in Table 1 recalculated based on the unified metric (sorted by "Performance+Potential"):
>
> | Algorithm | Potential | Performance+Potential |
> |-----------|-----------|----------------------|
> | MPC       | 0.0170    | 1.0862              |
> | BOLA      | 0.0250    | 1.0400              |
> | HYB       | 0.0679    | 0.9859              |
> | BBA       | 0.0181    | 0.7724              |
> | Pitree    | 0.0329    | 0.3407              |
>
> We can still observe that: 1. The HYB algorithm based on both buffer and bandwidth has higher potential than BOLA and BBA which are solely buffer-based; 2. The HYB algorithm with simple single-step logic has higher potential than the multi-step decision-making MPC.
>
> ## **2. Real-World Validation with Dash.js**
>
> As a solution to address the pain points of algorithm deployment in real environments, we agree that adding real environment experiments would make Crucible more convincing. To address the limitations of simulated environments, we conducted real network environment validation using Dash.js. We deployed Dash.js on the client side, connected to cloud servers via public WiFi, and tested five heuristic algorithms (MPC, BOLA, BBA, Pitree, HYB) as well as the reinforcement learning-based Pensieve as a control. Each algorithm controlled 3 minutes of real playback of Envivio video. Crucible performed optimization based on real playback logs, configured for 3 iterations and a maximum of 10 Bayesian iterations. The results are shown in the following table:
>
> | Algorithm    | Original | Crucible-Tuned |
> |--------------|----------|----------------|
> | HYB          | 1.40     | 1.72           |
> | BBA          | 1.56     | 1.72           |
> | BOLA         | 1.20     | 1.54           |
> | Pitree       | 1.73     | 1.73           |
> | MPC          | 1.72     | 1.79           |
> | Pensieve(RL) | 1.66     | -              |
>
> The real-world results validate two findings: (1) tuned simple heuristic algorithms (HYB, BBA) can outperform black-box algorithms; (2) algorithm potential correlates with its expressiveness and interpretability, with MPC achieving the highest post-tuning performance while complex Pitree showed no performance change after tuning.
>
> ## **3. Multi-Level Expert Simulation**
>
> Multi-level expert simulation is primarily based on Crucible's workflow. The capability stratification is based on computational resources consumed during the tuning process, rather than prompt design. Specifically, this includes:
>
> - **Reflection iterations**: Number of iterative optimization rounds by the LLM agent
> - **Bayesian optimization calls**: Frequency of traditional optimization tool invocations
>
> We only define interaction interfaces and basic knowledge in prompts (input/output formats, optimization objectives, etc. The specific prompt design can be found in Crucible/abr/server.py and Crucible/scheduling/server.py in the supplementary materials), and analyze performance improvements under different resource consumption levels (different iteration counts, different Bayesian optimization frequencies) through Figures 3 and 4 in sections 4.2 and 4.3.
>
> ## **4. Comparison with Existing Optimization Methods**
>
> **Comparison with Tuning Methods**
>
> Crucible treats other optimization methods (such as Bayesian optimization) as tool calls. The relative improvement results from section 4.2 and Figure 3 comparing with Bayesian optimization demonstrate: (1) Crucible transcends the limitations of traditional parameter optimization and achieves higher post-adjustment results through logic adjustment; (2) traditional optimization tools can enhance Crucible's tuning effectiveness. Comparing Figure 3(c) with Figure 3(a), when Bayesian optimization becomes available, the effectiveness rate of control logic adjustments made by LLM improved from approximately 40% to 60% compared to the same number of Bayesian optimization iterations. This indicates that in about 20% of scenarios, the general direction of control logic adjustments made by LLM was correct, but the specific parameter selection was problematic. Therefore, with Bayesian optimization, this portion of adjustments also became effective.
>
> **Comparison with Direct Optimization by Reinforcement Learning Methods**
>
> For black-box algorithms like RL, we employ decision tree conversion for potential analysis (such as Pitree in Table 1). Across the four datasets(Oboe, FCC, Puffer, 3G) in Table 1, Pensieve's average performance was 0.87, outperforming simple heuristic algorithms (BBA). However, after tuning, all four heuristic algorithms surpassed the RL algorithm, demonstrating the advantage of interpretable algorithms under tuning.

---

> > ### Author Response · Authors · 2025-08-06
> >
> > Dear Reviewer,
> >
> > Thank you for your valuable feedback. We appreciate it greatly. With only two days left in the rebuttal period, if you have any further comments, suggestions, or questions needing clarification, please feel free to share them.  We very appreciate your time and engagement.
> >
> > Best regards,
> >
> > Authors

---

> ### Comment · Reviewer_4H7x · 2025-08-07
>
> The authors' rebuttal provides a thoughtful and well-organized response that clarifies several key aspects of the paper. The effort to unify the environment distance metric through performance-based characterization is a meaningful step toward improving theoretical soundness, though a more formal treatment would be beneficial in the future. The inclusion of preliminary real-world validation using Dash.js adds practical value and supports the applicability of Crucible beyond simulations. The explanation of the multi-level expert simulation mechanism—grounded in computational resource usage rather than prompt design—clarifies the modeling choices and aligns with the workflow. Additionally, the expanded comparison with traditional optimization and RL-based methods helps better situate Crucible within the broader literature. While some concerns remain partially open, particularly regarding formalization and generalizability, the rebuttal has strengthened the overall presentation and reinforced the contribution's relevance. I maintain my original positive score.

---

> > ### Author Response · Authors · 2025-08-08
> >
> > Dear Reviewer,
> >
> > Thank you very much for your recognition of our response and for your valuable comments. Your precise feedback has greatly helped us improve the quality and clarity of our paper. We would like to take this opportunity to further elaborate on the core positioning and contributions of our work regarding the "formalization" and "generalizability" issues you mentioned.
> >
> > ## 1. About the Core Contribution: Defining and Quantifying a Practical Problem
> >
> > The core objective of our work is to propose and demonstrate a problem that is crucial in the real world but has not been sufficiently explored: the "Tuning Potential" of algorithms. We believe that leveraging LLMs provides us with an unprecedented opportunity to explore and quantify this "potential." The intention of this paper is not to provide a once-and-for-all ultimate answer for quantifying "potential," but rather to introduce this problem for the first time as an important, quantifiable, and assessable metric into the academic community's perspective.
> >
> > ## 2. About Generalizability
> >
> > To demonstrate the universality of the "tuning potential" concept and the effectiveness of our proposed Crucible framework, we carefully selected a series of experimental scenarios:
> >
> > Complex real-world systems: We initially chose ABR and scheduling algorithms because they are common and impactful problems in real systems.
> > Responding to your suggestions: Following your valuable recommendations, we added real network environment experiments based on Dash.js, which directly validated the applicability of our method in non-simulated environments.
> > Classic control problems: Meanwhile, to further address concerns about generalizability, we introduced Cart-Pole, a classic control scenario.
> >
> > This series of experiments spans from classic, theoretically-oriented control tasks to complex, engineering-driven real-world applications. We believe this design has powerfully demonstrated the universal importance of "tuning potential" as a concept, as well as the broad applicability of our Crucible framework in evaluating this potential across different domains.
> >
> > ## 3. About Formalization
> >
> > We completely agree that a more complete and rigorous formal definition will be an important direction for future work in this field. However, we believe that the greatest contribution of this paper lies in "pioneering" rather than "concluding."
> >
> > Crucible as a pioneering framework: Crucible is the first systematic framework we propose to address the new problem of "quantifying tuning potential." Its design, including performance-oriented environment distance metrics and multi-level expert simulation mechanisms, represents meaningful attempts to solve this practical problem.
> >
> > Value as a catalyst: We believe that better designs will certainly emerge in the future. Perhaps more sophisticated distance formulas, more efficient optimization tools, or methods that align more deeply with human domain knowledge. We eagerly anticipate seeing these advances and believe our work serves as the starting point and catalyst for all such discussions and innovations. If our paper can inspire broader interest and deeper research in the community regarding "how to better quantify algorithmic potential," we consider the value of this paper to have been realized.

---

### Note · Authors · 2025-08-11

Dear ACs and Reviewers,

We sincerely thank you and all the reviewers for your valuable time and insightful feedback during the review process. You have provided invaluable suggestions for this paper, which explores a new optimization direction in the traditional field of control algorithms. Your guidance on how to improve its presentation and more clearly address this problem—one that is highly practical in industry but under-explored in academia—has been instrumental in benefiting the broader community.

In response to your valuable feedback, we have made relevant enhancements in three aspects:

**Expanded Experimental Validation:** As the reviewers' suggestions, we have now added experiments on a classic control task (Cart-Pole) and a real-world system (Dash.js). This achieves comprehensive coverage—from classic control tasks and algorithms that are widely applied in production systems, to real-world systems—making the paper's experimental evaluation more thorough.

**Strengthened Theoretical Foundation:** We have formalized the environmental distance metric, which was originally used for scheduling algorithms, into a unified formula. This reduces the effort of devising task-specific metrics and makes it easier to apply Crucible to a wider range of control algorithms.

**Enhanced Robustness Analysis:** We have added a comparative analysis across different mainstream LLMs (e.g., GPT-4o-mini, Claude series) to demonstrate that while the absolute potential scores generated by Crucible may vary with the specific LLM, the overall relative rankings among algorithms remain largely stable.

We are very pleased to see that the reviewers who continued to engage in the discussion (4H7x, BJf8) have acknowledged our revisions.  BJf8 stated that their initial concerns were "satisfactorily addressed," and Reviewer 4H7x also noted that our work has been strengthened. At the same time, we have provided detailed, point-by-point responses and supplementary experiments for the valuable suggestions from all reviewers. We understand that everyone is very busy and sincerely hope that our responses can address any remaining concerns you may have.

As we stated in our response to 4H7x, our work is primarily about: "Defining and Quantifying a Practical Problem." Inspiring future research on algorithmic potential would be this paper's greatest success.

Finally, we sincerely thank the ACs and the four reviewers for reviewing this paper.

---

### Decision · Program_Chairs · 2025-09-17

**Decision:**

Accept (poster)

**Comment:**

The paper presents a framework for tuning the parameters of a control algorithm using an LLM, who plays the role of a domain expert. Using adaptive bitrate/scheduling algorithms as applications, the paper demonstrates improved performance over several reference scheduling algorithms.

During the rebuttal phase, the added a comparative analysis across different mainstream LLMs (e.g., GPT-4o-mini, Claude series) and expanded to a classic control task (Cart-Pole) and a preliminary real-world validation using Dash.js. It would be good to include both in the paper.

That said, more broadly, the paper would be strengthened by the application of the proposed algorithm to more domains beyond communications. As one reviewer noted, the paper demonstrates Crucible’s applicability to ABR and scheduling but does not discuss the limitations of extending the framework to other domains (e.g., robotics, real-time systems) where control algorithms may have different constraints or data distributions. A concerned remained that the task-specific environment distance metrics (Wasserstein for ABR, ratio-based for scheduling) may not generalize well.

Reviewers generally wanted to see more details regarding the implementation and experimental setting. These included a description of prompts used, but also formal/precise definitions of distance metrics employed. The authors have addressed these concerns in the rebuttal, and these edits should also make it to the paper.